# MDC1 PST-repeat region promotes histone H2AX-independent chromatin association and DNA damage tolerance

Israel Salguero [1], Rimma Belotserkovskaya [1,4], Julia Coates[1,4], Matylda Sczaniecka-Clift[1], Mukerrem Demir[1], Satpal Jhujh [1,3], Marcus D. Wilson [2] & Stephen P. Jackson [1]*

Histone H2AX and MDC1 are key DNA repair and DNA-damage signalling proteins. When DNA double-strand breaks (DSBs) occur, H2AX is phosphorylated and then recruits MDC1, which in turn serves as a docking platform to promote the localization of other factors, including 53BP1, to DSB sites. Here, by using CRISPR-Cas9 engineered human cell lines, we identify a hitherto unknown, H2AX-independent, function of MDC1 mediated by its PST-repeat region. We show that the PST-repeat region directly interacts with chromatin via the nucleosome acidic patch and mediates DNA damage-independent association of MDC1 with chromatin. We find that this region is largely functionally dispensable when the canonical γH2AX-MDC1 pathway is operative but becomes critical for 53BP1 recruitment to DNA-damage sites and cell survival following DSB induction when H2AX is not available. Consequently, our results suggest a role for MDC1 in activating the DDR in areas of the genome lacking or depleted of H2AX.

[1] Wellcome Trust/CRUK Gurdon Institute and Department of Biochemistry, University of Cambridge, Tennis Court Road, Cambridge CB2 1QN, UK. [2] Wellcome Centre for Cell Biology, University of Edinburgh, Michael Swann Building, Kings Buildings, Mayfield Road, Edinburgh EH9 3JR, UK. [3] Present address: Institute of Cancer and Genomic Sciences, University of Birmingham, Edgbaston, Birmingham B15 2TT, UK. [4] These authors contributed equally: Rimma Belotserkovskaya, Julia Coates.  *email: s.jackson@gurdon.cam.ac.uk

Cells are constantly subjected to a plethora of exogeneous and endogenously-derived DNA damaging agents. Among the different kinds of DNA damage, double-strand breaks (DSBs) are considered to be the most dangerous lesions, as they can trigger cell death, mutations and genome rearrangements, and can contribute to the development of cancer[1–3]. To protect the genome from DSBs, cells have evolved various proteins that are recruited to damaged chromatin regions to engage DNA repair processes and to trigger a signalling cascade that, amongst other things, can induce cell death or temporary or permanent delays in cell cycle progression. Collectively, these DNA repair and associated signalling events and outcomes can be referred to as the DNA damage response (DDR)[1,2].

In most current models, DSB signalling is initiated by the MRE11-RAD50-NBS1 (MRN) complex, which senses and binds to DSB regions, and then recruits and activates the protein kinase, ATM (Fig. 1a; ref. [4]). Next, ATM phosphorylates Ser-139 of histone H2AX in the chromatin surrounding the DSB site, with this phosphorylated H2AX (γH2AX) then creating a docking site for the tandem BRCT domain of the DDR mediator protein MDC1[5]. Because MDC1 is constitutively bound to NBS1 via its phosphorylated SDTD region[6–9], it draws in more MRN-ATM complex to the vicinity of the DSB, thereby contributing to amplification of ATM recruitment and activation, and to spreading of γH2AX-MDC1 on adjacent chromatin. Furthermore, upon recruitment to DSB-associated chromatin regions, MDC1 is phosphorylated on a set of TQXF motifs by ATM, thereby creating docking sites that recruit the ubiquitin E3 ligase RNF8[10,11], which then ubiquitylates proteins in the vicinity and thereby triggers recruitment of another E3 ligase, RNF168 (there is currently debate over whether the relevant protein targeted by RNF8 is histone H1[12] or L3MBTL2[13]). Once RNF168 is localised to the DSB region, it ubiquitylates histone H2A, leading to recruitment and retention of DNA repair factors such as 53BP1[14–17] and its downstream effectors including RIF1, PTIP and the recently identified Shieldin complex[18]. Notably, it seems that the same or similar mechanisms leading to 53BP1 accrual at ionising radiation (IR) induced foci (IRIF) also mediate 53BP1 recruitment to assemblies called nuclear bodies (NBs) in G1 phase cells. These NBs appear to represent sites of DNA/chromatin damage arising when cells progress through mitosis in the presence of unreplicated DNA regions, as evidenced by their numbers being elevated when cells are treated with the DNA-replication inhibitor aphidicolin (APH)[19–21]. Significantly, while the full DSB signalling cascade happens in interphase cells, if mitotic cells sustain DSBs, the process is blocked at the stage of RNF8 recruitment[22–25]. This is achieved by mitotic phosphorylations in RNF8 and 53BP1 that inhibit their recruitment to MDC1 and H2A, respectively, and thereby prevent formation of telomere fusions and consequent chromosome missegregations[26]. Nevertheless, recruitment of MDC1 to DSBs in mitosis seems to be important to maintain genome stability, with recent work showing how this, at least in part, relies on MDC1 interacting with TOPBP1. It has been proposed that in these circumstances, TOPBP1 and MDC1 may play a bridging role to keep the two DSB ends in close proximity to facilitate their repair in the ensuing G1 phase[27].

In line with the γH2AX-MDC1 interaction being fundamental for accumulation of DSB repair and signalling factors on chromatin in the vicinity of DSBs, H2AX and MDC1 knockout mice display similar phenotypes: growth retardation, male infertility, immune defects, chromosome instability and IR hypersensitivity at both the organism and cellular levels[28,29]. Moreover, human cells lacking either MDC1 or H2AX exhibit defects in DSB signalling and DNA-damage checkpoint activation[30–33]. However, while deletion of the gene for H2AX (H2afx) in mice was

reported to enhance tumour formation only in a p53-null background[34,35], Mdc1 knockout mice were reported to display a higher frequency of tumours even in the presence of p53 function[30]. These observations raise the possibility that there might be an additional, H2AX-independent function(s) for MDC1.

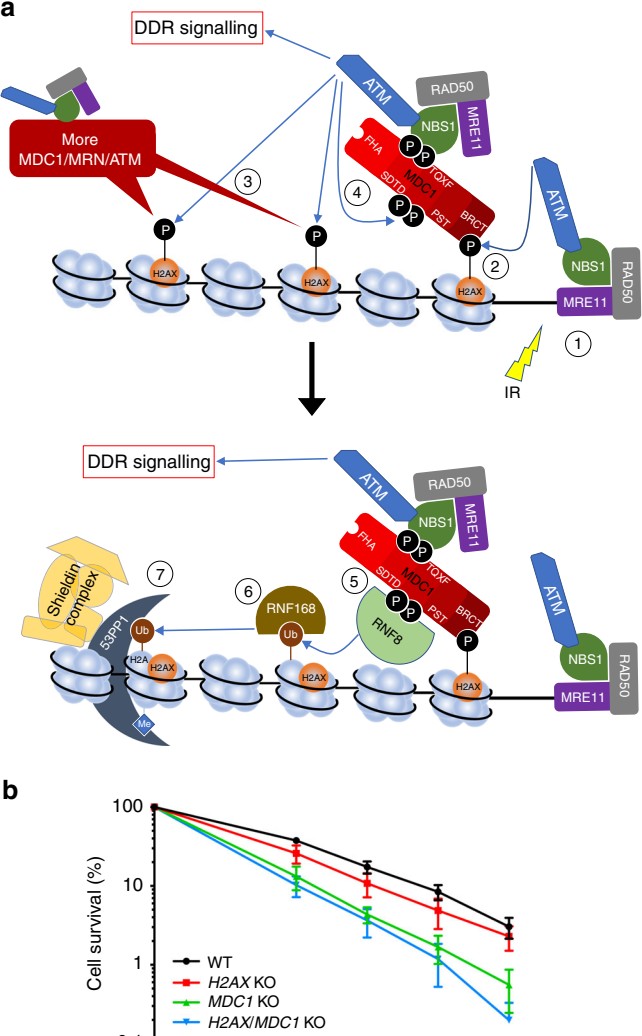

**Fig. 1** MDC1 loss causes greater IR sensitivity than H2AX loss. **a** Diagram depicting the main events in the signal transduction pathway that leads to 53BP1 accumulation on chromatin at DNA-damage sites: (1) DSB induction and MRN/ATM recruitment/activation. (2) ATM phosphorylates H2AX and this is recognised by MDC1, which brings in more MRN and ATM. (3) MDC1-mediated accumulation of ATM results in amplification of the γH2AX signal and, consequently, further recruitment of MDC1/MRN/ATM. (4) ATM also phosphorylates the TQXF cluster of MDC1. (5) Phosphorylated TQXF motifs are bound by RNF8, which ubiquitylates another protein(s). (6) This ubiuiqitylated protein(s) serves as a docking site(s) for RNF168. (7) RNF168 ubiquitylates H2A/H2AX and this, together with constitutive histone H4-K20-methylation, creates a platform that recruits 53BP1 and the Shieldin complex. **b** Clonogenic survival assays towards ionising radiation (IR) in specified RPE-1 genetic backgrounds. The tendency for $MDC1^{-/-}$ $H2AX^{-/-}$ double knockout cells to be slightly more IR sensitive than $MDC1^{-/-}$ single knockout cells might be explained by 53BP1 binding γH2AX in a MDC1-independent fashion[37,38,57] and/or by replication stress caused by the lack of H2AX[33] ; $n = 6$/genotype (except for MDC1 KO $n = 4$); error bars s.e.m. Additional supporting data, including validation, genotyping and cell cycle profiling of knockouts, are presented in Supplementary Fig. 1

Here, by generating and characterising human cells precisely deleted for the *MDC1* and/or *H2AFX* (hereafter *H2AX*) genes, we show that MDC1 and H2AX are not equivalent in their ability to convey IR resistance. Furthermore, we document that MDC1 is able to promote the recruitment of 53BP1 to DSB sites in a γH2AX-independent manner. This ability requires the proline-serine-threonine rich PST region of MDC1 whose DDR role has hitherto been unclear. We also show that this PST-repeat region binds to nucleosomes, and thereby mediates constitutive association of MDC1 with chromatin, a function that becomes critical for IR resistance in the absence of H2AX.

## Results

**MDC1 enhances IR survival even in the absence of H2AX.** While the current model for recruitment of DDR proteins to DSB sites (Fig. 1a) implies that loss of H2AX or MDC1 should be functionally equivalent, no direct comparison has to our knowledge been reported. To address this issue, we used CRISPR-Cas9 genome engineering to create both single- and double-knockouts for the genes for these factors in the otherwise isogenic background of non-transformed human RPE-1 hTERT cells (Supplementary Fig. 1a, b). After first observing that none of these mutant cell lines presented substantial alterations in their cell cycle distributions or S-phase progression (Supplementary Fig. 1c), we tested the sensitivities of various cell clones to IR. Perhaps surprisingly, while only very mild hypersensitivity was observed in the case of *H2AX*⁻/⁻ cells, considerably more pronounced IR hypersensitivity was exhibited by both *MDC1*⁻/⁻ single knockout and *MDC1*⁻/⁻ *H2AX*⁻/⁻ double knockout cells (Fig. 1b; Supplementary Fig. 1d). We thus concluded that, contrary to our expectations, MDC1 must have a DDR function that is independent of its interaction with histone H2AX.

To gain insights into the mechanism(s) underlying the differences in IR sensitivity between the *H2AX*⁻/⁻ and the *MDC1*⁻/⁻ knockout cells, we first examined IR-induced phosphorylation events on DNA-PKcs, KAP1 and CHK2 (Supplementary Fig. 1e). This analysis revealed no overt differences between the *H2AX*⁻/⁻, *MDC1*⁻/⁻ and *H2AX*⁻/⁻*MDC1*⁻/⁻ genetic backgrounds, suggesting that the IR hypersensitivity of *MDC1* mutant cell lines was not caused by major defects in the phosphorylation cascade induced by IR.

**H2AX-independent effects of MDC1 on 53BP1 DNA-damage accrual.** In light of our findings and because MDC1 is known to be crucial for 53BP1 recruitment to DNA damage regions, we noted that previous reports have documented H2AX-independent recruitment of 53BP1 to DNA-damage sites[33,36]. Indeed, we found that 53BP1 accumulation in NBs was highly effective in the absence of H2AX (Fig. 2a, b; APH). Nevertheless, although the proportion of *H2AX*⁻/⁻ cells containing NBs was similar to that of wild-type cells, the number NBs per cell was lower in the *H2AX*⁻/⁻ background (Supplementary Fig. 2a). Given that neither the size nor the staining intensity of 53BP1 NBs seemed to be decreased by the lack of H2AX, the lower number of NBs per cell in the absence of H2AX could reflect the existence of different types of lesions generating NBs, with some but not other types being amenable to H2AX-independent 53BP1 accumulation. Notably, while 53BP1 IRIF formation was reduced by H2AX inactivation, IRIF still clearly formed in some *H2AX*⁻/⁻ cells (Fig. 2a, b; IR; Supplementary Fig. 2a, bottom panel). Although we do not have a full explanation for the differential effects of H2AX loss on NBs and IRIF, we note that H2AX-independent IRIF frequently occur in G1 cells (Supplementary Fig. 2b), the cell cycle stage in which NBs are evident. It may thus be that G1 cells more easily mediate 53BP1 accumulation and/or

retention in the absence of H2AX than do cells in other cell-cycle stages. Alternatively, the distinct nature of the underlying lesions in 53BP1 IRIF and 53BP1 NBs—DSBs generated directly by IR versus DSBs arising during mitosis in unreplicated DNA regions—could account for the differences observed. Most crucially, we found that unlike the situation in response to H2AX loss, localisation of 53BP1 to both NBs and IRIF was strongly diminished by MDC1 loss (Fig. 2a, b; Supplementary Fig. 2a; the residual 53BP1 recruitment to NBs in *MDC1*⁻/⁻ cells might reflect the ability of 53BP1 to bind γH2AX directly[37,38]). Furthermore, we observed that 53BP1 NBs and residual IRIF in H2AX-deficient cells were totally abolished by MDC1 inactivation (Fig. 2a, b; Supplementary Fig. 2a).

**H2AX-independent association of MDC1 with NBs requires 53BP1.** In light of our findings, we assessed for enrichment of MDC1 at sites of DNA damage in either the presence or absence of H2AX. While no MDC1 IRIF were observed in *H2AX*⁻/⁻ cells, clear accumulation of MDC1 in NBs was detected in this setting (Fig. 2c). To explain the different responses in NBs and IRIF, we speculate that the features of NBs that allow recruitment of large amounts of 53BP1[19,20] may facilitate the accumulation of MDC1 even in *H2AX*⁻/⁻ cells. Supporting this hypothesis, it has been reported that MDC1 and 53BP1 can directly interact in a manner mediated by the tandem BRCT region of MDC1[39]. To assess if this interaction might be responsible for the MDC1 enrichment at NBs in *H2AX*⁻/⁻ cells, we used small interfering RNA (siRNA) treatments to deplete 53BP1 from such cells and control *H2AX*⁺/⁺ cells. While 53BP1 depletion did not produce any apparent effect on MDC1 NB association in *H2AX*⁺/⁺ settings, it markedly reduced the frequency, and especially the extent, of MDC1 accumulation at NBs in *H2AX*⁻/⁻ cells (Fig. 2d; Supplementary Fig. 2c, d). Similar effects were observed following siRNA depletion of RNF8, which is required for 53BP1 accrual at DNA-damage sites[10,11]. By contrast, depleting either of two components of the Shieldin complex, SHLD1 and SHLD2, did not markedly affect NB formation by MDC1 in *H2AX*⁻/⁻ cells (Fig. 2d; Supplementary Fig. 2c, d). Collectively, these data indicated that 53BP1, but not SHLD1/2, is crucial for MDC1 recruitment and/or retention at NBs specifically in H2AX-deficient cells.

**The MDC1 PST region promotes the DDR in the absence of H2AX.** Having established that localisation of 53BP1 to DNA damage sites depends on MDC1 in cells lacking H2AX, we next examined which of the structural and functional domains of MDC1 are needed for this (Fig. 3a)[40]. Thus, we complemented *MDC1*⁻/⁻ single and *MDC1*⁻/⁻ *H2AX*⁻/⁻ double knockout cells with green-fluorescent protein (GFP)-tagged wild-type or mutant versions of *MDC1* individually lacking each of the domains, then assessed 53BP1 accumulation at NBs arising after aphidicolin treatment. Deleting the MDC1 SDTD region only produced a small reduction in the number of NBs per cell in the *H2AX*⁻/⁻ mutant background (Supplementary Fig. 3a, b) and no detectable effect on the percentage of cells containing 53BP1 NBs (Fig. 3b). By contrast, mutating Thr residues of the MDC1 TQXF cluster to Ala (AQXF) almost completely abrogated 53BP1 localisation to NBs in a manner that was not dependent on H2AX status (Fig. 3b; Supplementary Fig. 3a, b). This effect was as expected because the Thr residues in the TQXF motifs are phosphorylated by ATM to generate binding sites for RNF8, which is crucial for effective 53BP1 recruitment[10,11]. We also found that deleting the MDC1 tandem BRCT domain reduced 53BP1 NB formation independently of *H2AX*⁻/⁻ status (Fig. 3b; Supplementary Fig. 3a, b). We speculate that this effect likely reflects the MDC1

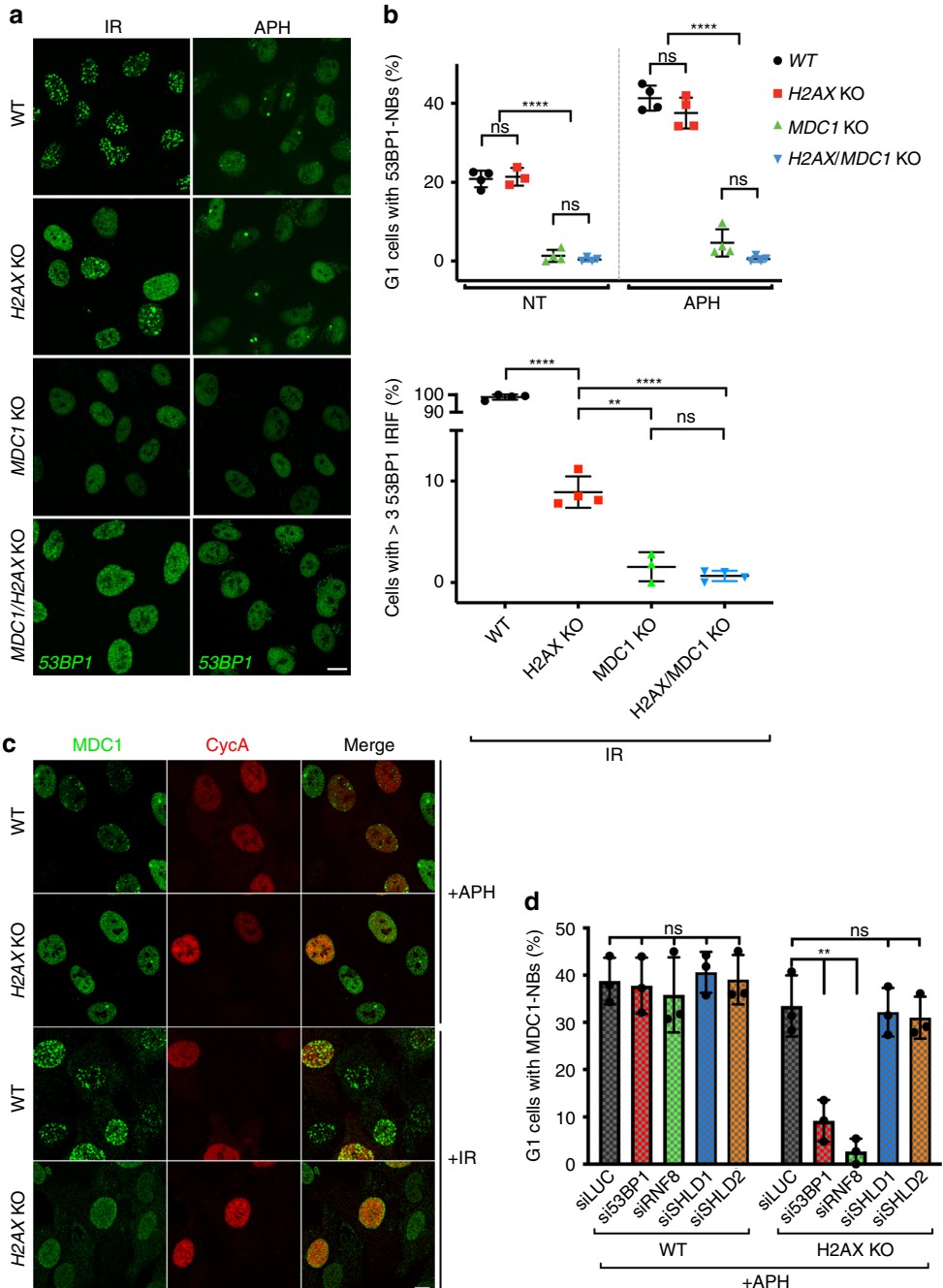

**Fig. 2** 53BP1 localisation to DNA-damage sites in $H2AX^{-/-}$ cells depends on MDC1. **a** Representative immunofluorescence images of 53BP1 NB formation after 24 h of 0.4 μM aphidicolin (APH) treatment, and of 53BP1 IRIF 1 h after IR (3 Gy) exposure in wild-type RPE-1 and knockout cell lines. **b** Quantification of 53BP1-NBs and 53BP1 IRIF in cells treated as in **a**. Cyclin A staining was used to differentiate G1 from S/G2 cells; n = 4/genotype (except for non-treated $H2AX$ KO and IR $MDC1$ KO n = 3); error bars s.e.m.; ****p < 0.0001; two-tailed Student's t test. **c** Representative immunofluorescence images showing MDC1 localisation at NBs or IRIF after APH or IR treatments (as in **a**) in the RPE-1 $H2AX^{+/+}$ and $H2AX^{-/-}$ cell lines. Cyclin A staining was used to distinguish between G1 and S/G2 cells. **d** Quantifications showing MDC1 localisation at NBs in the RPE-1 $H2AX^{+/+}$ and $H2AX^{-/-}$ cell lines after 24 h treatment with 0.4 μM APH. Cells were depleted for 53BP1, RNF8, SHLD1 or SHLD2 by siRNA for 48 h before the APH treatment; n = 3/genotype; error bars s.e.m.; **p < 0.01; two-tailed Student's t test. Supporting data, including quantifications of number of NBs and IRIF per cell and validation of 53BP1, RNF8, SHLD1 and SHLD2 depletion are shown in Supplementary Fig. 2. Scale bars, 10 μm

BRCT region not only binding γH2AX, but also recruiting 53BP1 to NBs via a direct interaction with 53BP1[39]. In addition, we observed that deleting the MDC1 FHA domain reduced 53BP1 NB formation in both $H2AX^{+/+}$ and $H2AX^{-/-}$ cells (Fig. 3b; Supplementary Fig. 3a, b). Previous work has shown that the MDC1 FHA domain promotes MDC1 oligomerization and

thereby potentiates IRIF formation by it and other DDR factors such as 53BP1[41,42]. Our results suggest that a similar mechanism may operate in NBs.

Contrasting with the impacts documented above were our findings relating to the MDC1 proline-serine-threonine rich (PST) repeat region, which in human cells comprises 13 imperfect

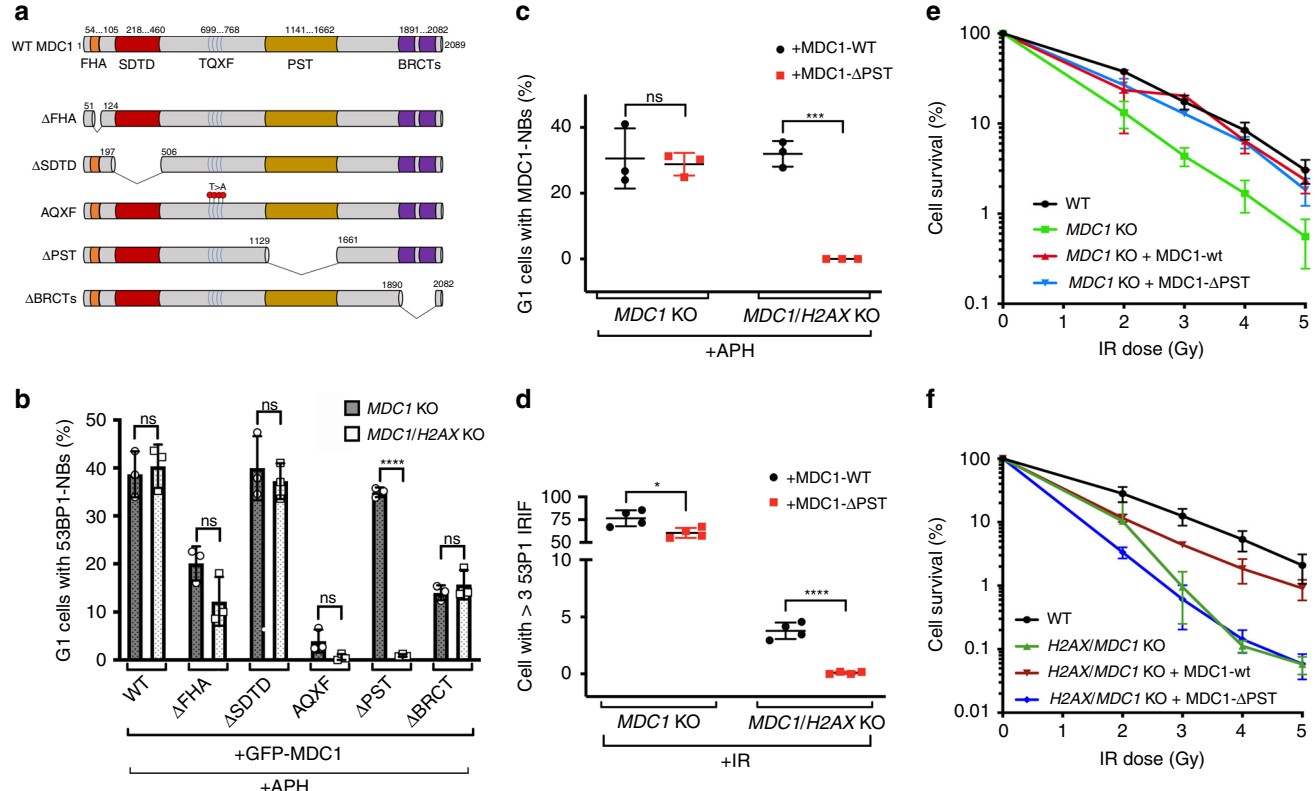

**Fig. 3** The PST region of MDC1 mediates key DDR functions in cells lacking H2AX. **a** Schematic representation of MDC1 architecture and the different mutant versions used in this work. **b** Quantification of 53BP1-NB formation after 24 h of 0.4 μM APH treatment in RPE-1 $MDC1^{-/-}$ $H2AX^{+/+}$ and $MDC1^{-/-}$ $H2AX^{-/-}$ cells complemented with indicated mutant versions of GFP-tagged MDC1; $n = 3$/genotype; error bars s.e.m.; ****$p < 0.0001$; two-tailed Student's $t$ test. **c** Quantification of MDC1-NB formation after APH treatment (as in **b**) of RPE-1 $MDC1^{-/-}$ $H2AX^{+/+}$ and $MDC1^{-/-}$ $H2AX^{-/-}$ cells complemented either with the WT or ΔPST versions of GFP-MDC1; $n = 3$/genotype; error bars s.e.m.; ***$p < 0.001$; two-tailed Student's $t$ test. **d** Quantification of 53BP1 IRIF 1 h after IR (3 Gy) treatment of RPE-1 $MDC1^{-/-}$ $H2AX^{+/+}$ and $MDC1^{-/-}$ $H2AX^{-/-}$ cells complemented with WT or ΔPST versions of GFP-MDC1; $n = 4$/genotype; error bars s.e.m.; ****$p < 0.0001$, *$p < 0.05$; two-tailed Student's $t$ test**. e** Clonogenic survivals after IR treatments of RPE-1 $MDC1^{-/-}$ cells complemented with WT or ΔPST versions of GFP-MDC1; $n = 5$/genotype; error bars s.e.m. **f** Same as in (**e**) but for RPE-1 $MDC1^{-/-}$ $H2AX^{-/-}$ cells; $n = 3$/genotype; error bars s.e.m. Supporting information, as representative images and quantifications of number of 53BP1 foci per cell are presented in Supplementary Fig. 3

copies of an ~40 amino-acid-residue motif. Strikingly, deleting this PST-repeat region had a major effect on 53BP1 NB formation only in the absence of H2AX. Thus, while expression of MDC1-ΔPST in $MDC1^{-/-}$ cells restored 53BP1 NB formation to near normal levels, 53BP1 NBs were undetectable when MDC1-ΔPST was expressed in $MDC1^{-/-}$ $H2AX^{-/-}$ cells (Fig. 3b; Supplementary Fig. 3a, b). In line with these observations, we found that deleting the MDC1 PST-repeat region did not affect the accumulation of MDC1 in NBs in otherwise wild-type cells, yet totally abrogated MDC1 NB accrual in the absence of H2AX (Fig. 3c; Supplementary Fig. 3c). As we had established that residual 53BP1 IRIF formation in $H2AX^{-/-}$ cells requires MDC1 (Fig. 2a, b; Supplementary Fig. 2a, b), we assessed whether this required the MDC1 PST-repeat region. Indeed, in accord with our other data, expression of wild-type MDC1 but not MDC1-ΔPST in the $MDC1^{-/-}$ $H2AX^{-/-}$ knockout background allowed 53BP1 IRIF formation in a subset of cells (Fig. 3d; Supplementary Fig. 3d, e; as observed in the case of NBs, deleting the PST-repeat region had no or only a mild effect in $H2AX^{+/+}$ cells).

To address the potential biological importance of the MDC1 PST-repeat region in the $H2AX^{-/-}$ genetic background, we assessed the impact of deleting this section of MDC1 on clonogenic cell survival in response to IR exposure in both H2AX-proficient and H2AX-deficient settings. As shown in Fig. 3e, expression of wild-type MDC1 or MDC1-ΔPST

essentially fully alleviated the IR hypersensitivity of $H2AX^{+/+}$ $MDC^{-/-}$ knockout cells. In stark contrast, while wild-type MDC1 largely alleviated the IR hypersensitivity of $H2AX^{-/-}$ $MDC^{-/-}$ double mutant cells, MDC1-ΔPST did not (Fig. 3f). Taken together with our other findings, these results supported a model in which MDC1 PST-region mediated recruitment of proteins to DSBs promotes cell survival in response to IR when H2AX is absent.

**The PST region fosters constitutive MDC1 chromatin retention.** To explore the function(s) of the MDC1 PST region, we tested whether it was recruited to DSB regions in cells when fused to GFP either alone or together with the MDC1 BRCT domain. As shown in Supplementary Fig. 4a, in contrast to IRIF and NB formation by the GFP-PST-BRCT fragment, we were not able to detect localisation of the GFP-PST construct at these sites. We thus concluded that the function of the MDC1 PST-repeat region is not to specifically target MDC1 to sites of DNA damage. This is consistent with the fact that, although the accumulation of 53BP1 at sites of DNA damage is MDC1-dependent in the absence of H2AX, we could not detect any enrichment of MDC1 itself at IRIF in the $H2AX^{-/-}$ genetic background (Fig. 2c).

While MDC1 is reported to be constitutively associated with chromatin independently of DNA damage[43], to our knowledge, no

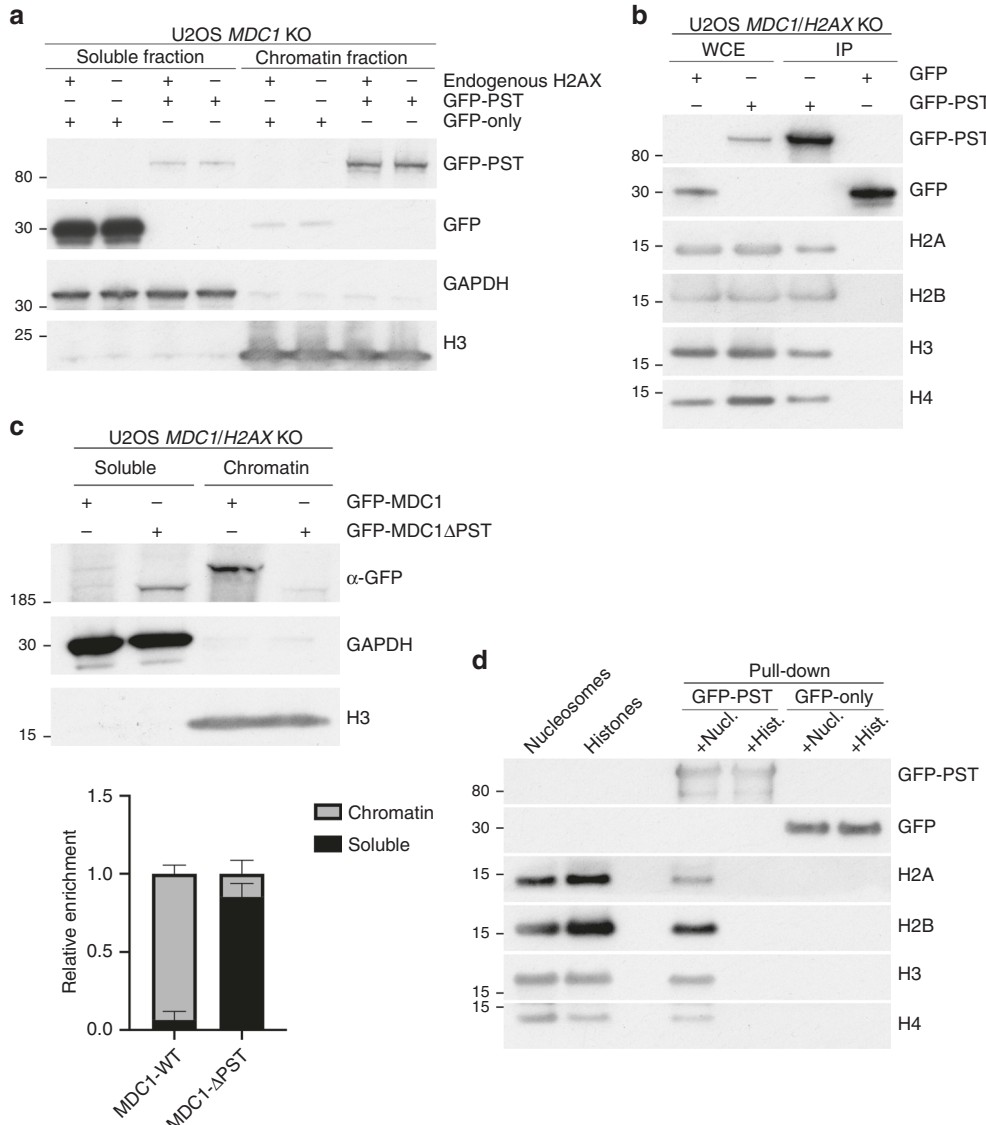

**Fig. 4** MDC1 PST region binds nucleosomes and promotes DNA damage-independent association of MDC1 with chromatin. **a** Chromatin fractionation of U2OS *MDC1*⁻/⁻ and *MDC1*⁻/⁻ *H2AX*⁻/⁻ cells transfected with plasmids expressing a GFP-PST construct or GFP-only. **b** GFP pulldowns from extracts of U2OS *MDC1*⁻/⁻ *H2AX*⁻/⁻ cells expressing GFP-PST or GFP-only were analysed by western blotting using antibodies against the four core histones and GFP. **c** Representative image of chromatin fractionation of U2OS *MDC1*⁻/⁻ *H2AX*⁻/⁻ cells transfected with plasmids expressing GFP-tagged full-length or ΔPST versions of MDC1 (top panel) and quantification of the relative MDC1 abundance in each fraction (lower panel); *n* = 4; error bars s.e.m. As previously reported[47], we found that GFP-MDC1ΔPST mutant protein was expressed in much higher amounts than wild-type GFP-MDC1, implying that the PST region promotes MDC1 turnover. To normalise protein levels, we harvested cells at 48 h after transfection for full-length MDC1 and at 8 h after transfection in the case of ΔPST-MDC1 (Supplementary Fig. 4f). **d** Western blot to detect the 4 core histones in samples derived from biochemical binding assays between GFP-PST or GFP-only (both purified from transfected HEK293 cells) and core histones purified from calf thymus or native mono-nucleosomes purified from HeLa cells. Additional data can be found in Supplementary Fig. 4, which includes GFP-PST protein localisation after DNA damage, validation of knockouts and PST-DNA binding assays

function has so far been established for this association. To evaluate whether the PST region might bind chromatin, we performed chromatin fractionation studies on *MDC1*⁻/⁻ *H2AX*⁺/⁺ and *MDC1*⁻/⁻ *H2AX*⁻/⁻ U2OS cells (Supplementary Fig. 4b–d) transfected with constructs expressing GFP alone or GFP fused to the MDC1 PST-repeat region. Strikingly, in contrast to GFP alone, the PST-GFP protein was enriched in the chromatin fraction in a manner that was independent of *H2AX* status (Fig. 4a). We noticed that, when overexpressed, the GFP-PST protein accumulated in areas of the nucleus resembling nucleoli (Supplementary Fig. 4a). However, it is unlikely that this localisation was responsible for the association of the GFP-PST fragment to chromatin fractions, as the

apparent nucleolar localisation was lost when cells were detergent pre-extracted before fixation (with a similar buffer to the one used for chromatin fractionation) while the rest of the nuclear GFP-PST remained resistant to pre-extraction (Supplementary Fig. 4e). In line with this conclusion, when we carried out immunoprecipitation studies with cell extracts expressing GFP or the GFP-PST fusion construct, the latter but not the former co-immunoprecipitated with all four core histones (Fig. 4b).

The above findings showed that the MDC1 PST-repeat region directly or indirectly binds chromatin. To see whether this region was necessary for chromatin association by MDC1, we transfected *H2AX*⁻/⁻ *MDC1*⁻/⁻ U2OS cells with plasmids expressing

GFP fused to full-length MDC1 (GFP-MDC1) or GFP fused to MDC1 bearing a deletion of the PST-repeat region (GFP-MDC1ΔPST). Notably, while wild-type MDC1 robustly associated with chromatin, this association was much less strong in the context of MDC1 lacking the PST-repeat region, where most of the protein remained in the soluble, non-chromatin fraction (Fig. 4c). Collectively, these results pointed to a hitherto unreported role for the MDC1 PST region in mediating DNA-damage independent association of MDC1 with chromatin.

**The MDC1 PST region interacts with the nucleosome acidic patch.** While our previous data had established that the MDC1 PST-repeat region binds chromatin, they did not show whether this binding was direct, or which histone or histone(s) might be involved in the interaction. To address these issues, we carried out in vitro binding assays with purified GFP-PST fusion protein derived from transfected HEK293 cells and commercial recombinant core histones expressed in and purified from *E. coli* and thereby lacking posttranslational modifications (PTMs). Notably, we did not detect binding of the MDC1 PST fragment to any of the recombinant histones (Supplementary Fig. 4g). We reasoned that the observed binding of the PST region to chromatin derived from cells but not recombinant histones could reflect indirect binding to chromatin via a factor that was not present in the recombinant histone preparation, chromatin binding requiring a histone PTM that was not present in the recombinant histones, or chromatin binding requiring more than one histone or even an intact nucleosome structure. To explore these possibilities, we carried out binding assays with histones purified from calf thymus, which carry PTMs, and with mono-nucleosomes purified from HeLa cells. Strikingly, the PST region did not bind any histone from the mammalian histone mix, but efficiently retrieved all four core histones from the preparation of mono-nucleosomes (Fig. 4d; Supplementary Fig. 4h). These data thus indicated that the MDC1 PST region binds chromatin directly, and furthermore suggested that this association is not mediated by a single histone but needs the whole nucleosome complex.

An obvious difference between the nucleosomes and the purified histones used in the above studies is the former but not the latter contain DNA that wraps around the histones to generate a stable histone octamer. However, we did not detect binding of the MDC1 PST region to sepharose beads bearing a DNA oligonucleotide (Supplementary Fig. 4i; note binding of the known DNA-binding protein PARP1 to beads bearing DNA but not native beads). These data suggested that PST-mediated MDC1 binding to nucleosomes and chromatin is unlikely to arise through DNA binding or, at least through DNA binding alone. Given that the PST region is positively charged (isoelectric point of 9.85; Isoelectric.org), we noted that several chromatin-interacting proteins recognise a negatively charged region of the nucleosome termed the acidic patch, formed by residues in both histone H2A and H2B[17,44,45]. We therefore explored whether the acidic patch was required for the PST-nucleosome interaction by performing binding assays with reconstituted recombinant nucleosomes. Thus, we observed that mutating key residues in the acidic patch dramatically reduced PST region binding to nucleosomes (Fig. 5a, b). These findings therefore highlighted roles for H2A-H2B acidic patch residues in mediating MDC1 PST-repeat region binding. Nevertheless, the PST fragment still had some residual binding to acidic-patch mutant nucleosomes (Fig. 5b), meaning that the MDC1-PST region might also mediate additional interactions with other surface(s) of the nucleosome.

All the chromatin/nucleosome-binding experiments described above were performed with the full-length PST region, composed of 13 degenerate copies of the ~40 amino-acid-residue repeat motif. We next investigated whether one repeat was enough to mediate nucleosome association or whether more than one repeat was needed for effective binding. To do this, we transfected HEK293 cells with constructs encoding GFP fused to the full-length (FL) MDC1 PST region (containing 13 repeats) or truncated GFP-fusion constructs containing 0, 1, 2 or 5 repeats. We then affinity-purified the proteins and tested them for their ability to bind nucleosomes in vitro. As shown in Fig. 5c, the protein containing 5 PST repeats bound purified mono-nucleosomes, although with lower efficiency than the full-length PST region, while undetectable binding was observed for GFP alone or the proteins containing 1 or 2 PST repeats. Complementary results were obtained when we carried out chromatin-fractionation studies with extracts of cells expressing GFP or the different GFP-PST derivatives: only the protein with 5 PST repeats bound chromatin detectably and this binding was less effective than exhibited by the full-length PST region (Fig. 5d). As discussed further below, these results suggest a cooperative binding model in which strong binding is brought about via multiple MDC1 PST repeats interacting with multiple acidic patches, and perhaps other nucleosome surfaces.

## Discussion

MDC1 is a key DDR protein, which functions by binding γH2AX and mediating localisation of the ubiquitin E3 ligases RNF8 and RNF168 to DNA damage sites, thereby promoting ubiquitylation of histone H2A and the ensuing recruitment of various DDR components, including 53BP1 and BRCA1. Here, we have shown that MDC1 functions in the DDR do not fully depend on its association with γH2AX, thus helping to explain the higher IR sensitivity of $MDC1^{-/-}$ knockout cells as compared to $H2AX^{-/-}$ cells. Our results also imply that MDC1 promotes survival to DNA damage in the absence of H2AX via its ability to help recruit repair factors to DSB regions. Furthermore, we have established that DNA-damage independent MDC1 association with chromatin is largely mediated by its PST-repeat region, a region that does not have any discernible impact on IR survival in a $H2AX^{+/+}$ background but becomes important for IR survival in $H2AX^{-/-}$ cells. These observations thus support a model in which chromatin binding by MDC1 mediates a DDR that minimises the toxicity of DNA damage when the canonical γH2AX-MDC1 axis is not available.

The PST-repeat region of MDC1 is conserved in vertebrates, although the number of repeats varies considerably between species (for example, 13 in human and 7 in mouse). It does not contain known structural or functional motifs and does not appear to display sequence homology to any other protein. Notably, previous studies have described DSB repair defects in cell lines carrying $\Delta PST$ mutant MDC1 alleles[46,47]. Furthermore, it was reported that the MDC1 PST region interacts with the Ku/DNA-PKcs (DNA-PK) complex and that this interaction is required for DNA-PKcs autophosphorylation and for DSB repair by non-homologous end joining (NHEJ), as measured by random plasmid integration assays[46]. However, we have been unable to observe any defect in the kinetics or extent of DNA-PKcs, CHK2 or KAP1 phosphorylation in our $MDC1^{-/-}$ or $MDC1^{-/-}$ $H2AX^{-/-}$ knockout cells as compared to wild-type controls. It was unexpected to observe no overt DDR defects in the absence of MDC1, although in support of our observations, a recent study similarly did not report signalling deficiencies in $MDC1^{-/-}$ U2OS cells[27]. Early literature on MDC1 reported contradictory effects of MDC1 depletion on the DDR signalling pathway, with some papers describing reduced phosphorylation of certain DDR factors and other papers showing a larger effect on other phosphorylations[30–32,48]. One explanation for the discrepancies

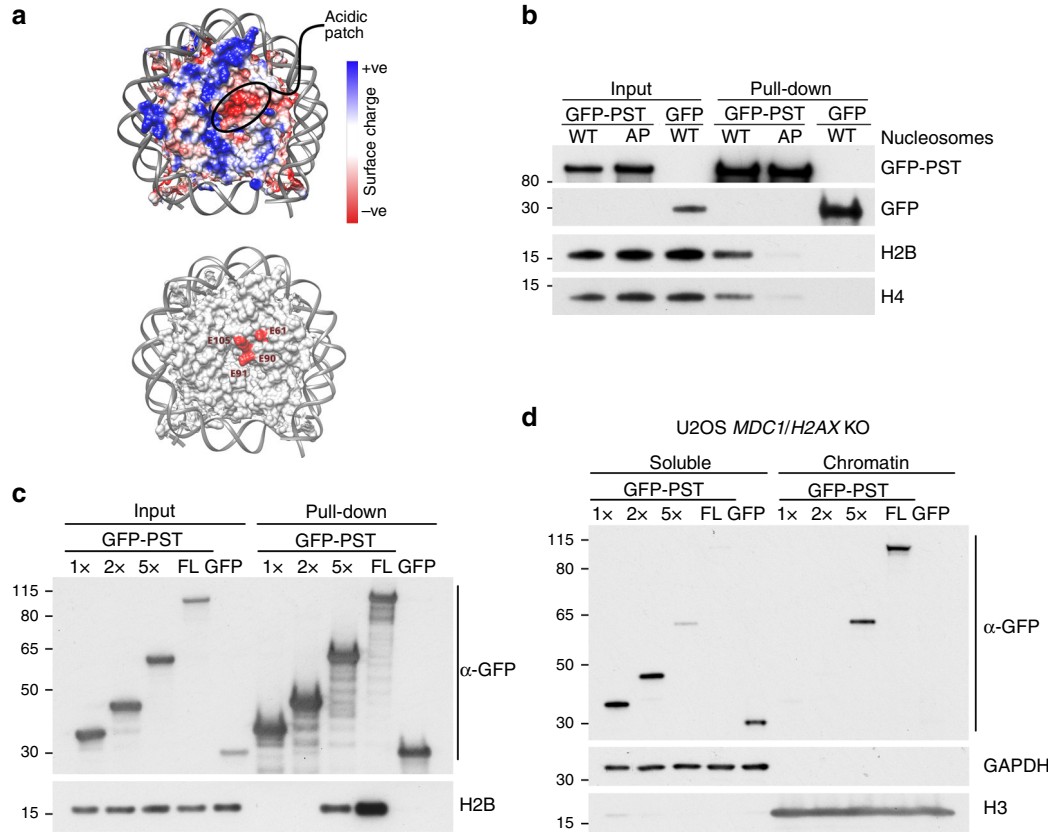

**Fig. 5** Binding of MDC1 PST region to chromatin is mediated by the nucleosome acidic patch and requires several PST repeats. **a** Depiction of the nucleosome and its acidic patch. Top: electrostatic potential view of the nucleosome surface, coloured according to coulombic surface charge. Bottom: view of the nucleosome surface and localisation of the mutated acidic patch residues. **b** Biochemical binding assay between GFP-PST or GFP-only and mono-nucleosomes reconstituted using recombinant histones. WT, all core histones are wild-type; AP, acidic patch mutant with the H2A E61/91/92 A and H2B E105A point mutations. **c** Biochemical binding assay between different GFP constructs, containing 0, 1, 2 or 5 repeats or the full-length (FL) PST fragment, and native mono-nucleosomes purified from HeLa cells. **d** Chromatin fractionation of U2OS $MDC1^{-/-}$ $H2AX^{-/-}$ cells transfected with plasmids expressing the indicated GFP-PST constructs or GFP-only

between our observations and some previous reports could be off-target effects of siRNAs used in the previous studies and/or the adaptation experienced by our cell lines after we had inactivated the *MDC1* gene. Furthermore, we have observed no discernible differences in DDR-factor phosphorylations between $MDC1^{-/-}$ $H2AX^{-/-}$ knockout cells complemented with *MDC1-WT* or *MDC1-ΔPST* constructs (Supplementary Fig. 5), implying that IR hypersensitivity observed in MDC1-deficient backgrounds is not caused by defects in the phosphorylation cascade triggered by DNA damage induction. In addition, we observed an effect of the *MDC1-ΔPST* allele on cell survival only in *H2AX*-deficient backgrounds, while DNA-PKcs/NHEJ defects are known to impact on cell survival when H2AX is present[49,50]. Collectively, our data point to chromatin binding by the MDC1 PST-repeat region as acting in an H2AX-independent manner to enhance cell survival via promoting the functions of 53BP1 and its downstream effectors at sites of DNA damage and perhaps the adjacent chromatin regions.

Mechanistically, we have shown that mutations in key residues in the nucleosome acidic patch, comprised by amino acids of H2A and H2B, severely impair the binding of the MDC1 PST-repeat region to nucleosomes in vitro. This result implies that the PST region interacts with the acidic patch of nucleosomes to promote chromatin association of MDC1 and associated factors in cells, a conclusion supported by our chromatin-fractionation data. It is notable that our work has shown more than two MDC1 PST repeats are needed to mediate effective binding to cellular

chromatin or to nucleosomes in vitro. While these observations are consistent with the fact that all vertebrate MDC1 homologues studied contain many PST repeats, they are puzzling because a nucleosome only contains two acidic patches. Our observations might therefore suggest that several repeats could interact cooperatively with the same acidic patch, or perhaps more plausibly, that certain PST repeats act as spacers to allow two non-adjacent PST repeats to interact simultaneously and cooperatively with two acidic patches on the opposing faces of a nucleosome. It is also tempting to speculate that the entire MDC1 PST region may make contacts with multiple adjacent nucleosomes in a way to change chromatin characteristics and influence the binding of other chromatin components. Clearly, more detailed biochemical, biophysical and structural studies will be needed to test these ideas and explore their potential impacts on chromatin organisation. In this regard, it is important to recognise that in addition to MDC1, several other proteins are known to interact with the acidic patch region of the nucleosome[51], although effective nucleosome binding by some of these, such as 53BP1 and RNF169, requires recognition of histone modifications in addition to the acidic patch itself[17,52]. In light of this and our findings, it will be interesting to examine if the H2AX-independent interaction between MDC1 and nucleosomes can be modulated by any histone posttranslational modification and/or whether MDC1 interactions with nucleosomes function synergistically or antagonistically with other nucleosome-binding proteins.

Taken together, our data suggest an important role for the MDC1 PST region in an H2AX-independent pathway to activate DDR factors at sites of DNA damage by a mechanism underpinned by MDC1 binding chromatin in a DNA-damage independent fashion. While it is difficult to envision how evolution could maintain such an alternative mechanism if it is relevant only in cells lacking H2AX, we note that the distribution of the H2AX variant histone along the genome is not homogeneous, with large areas of the human genome appearing to be largely or completely depleted of H2AX[53]. As we have shown that MDC1 constitutive binding to chromatin does not depend on H2AX, it is tempting to speculate that the importance of PST-repeat region mediated MDC1 chromatinization is to facilitate DDR activation when DSBs occur in such regions, with DSB-tethered MRN and ATM triggering MDC1 phosphorylations in the vicinity of the breaks and ensuing events in these settings. Further studies will be needed to test this and other hypotheses.

## Methods

**Cell culture**. RPE-1 FRT-derived cells were cultured in F12 (Ham's F12; Sigma-Aldrich) supplemented with 17 ml NaHCO3 7.5% per 500 ml (Sigma-Aldrich). U2OS-derived and HEK293 cells were cultured in Dulbecco's modified Eagle medium (DMEM; Sigma-Aldrich).

All media were supplemented with Tet-free 10% (vol/vol) fetal bovine serum (Life Technologies Ltd), 100 U ml[−1] penicillin, 100 μg ml[−1] streptomycin (Sigma-Aldrich) and 2 mM L-glutamine. For maintenance of the RPE-1 FRT-derived cells expressing GFP-tagged constructs, 0.5 mg ml[−1] G418 (Invitrogen) was used. All cells were originally obtained from the ATCC cell repository and routinely tested for mycoplasma.

**Generation of human stable cell lines and knockouts**. FRT-derived cells stably expressing inducible GFP-tagged constructs were generated by transfection of pcDNA5/FRT/TO-neo containing the GFP-tagged construct and pOG44 (1:4, respectively). Selection began after 48 h using 0.5 mg ml[−1] G418 (Invitrogen). *H2AX* knockout was generated in RPE-1 FRT cells, by co-transfection with an All-in-one plasmid[54], containing the gRNAs and the nickase Cas9 gene, and a donor plasmid with two *H2AX* homology arms flanking a cassette with *GFP* and puromycin-resistance genes. The sequences of the DNA oligonucleotides used to generate the guide-RNAs are listed in Supplementary Table 1. After puromycin selection, single-cell sorting by GFP expression was carried out using MoFlo (Beckman Coulter) to select for transfected cells. Single clones were expanded and screened by western blotting.

Deletion of *H2AX* in the RPE-1 FRT MDC1 null[54], U2OS or U2OS MDC1 null[54] cells was carried out by transfection of the same All-in-one plasmid, without using any donor plasmid. GFP-expressing cells were single-cell sorted, expanded and screened by western blotting.

*H2AX* DNA from candidate clones was amplified by PCR using the primers H2AFX Fwd and H2AFX Rev, and the PCR products were Topo-cloned and Sanger-sequenced using the primers H2AX seq1 Fw, H2AX seq1 Rv, H2AFX Fwd and H2AFX Rev (Supplementary Table 2).

**Plasmids**. Plasmids used to stably integrate the GFP-tagged constructs into FRT cells were generated as follows: The GFP-tagged MDC1 wild type and mutant versions were obtained by PCR from plasmids previously described in refs. [5,6,11]. PCR products were then cloned into the pcDNA5/FRT/TO-neo plasmid. In order to generate the GFP-PST and GFP-PST-BRCT expression vectors the corresponding fragments (1120–1667 for PST and 1120–2089 for PST-BRCT) were amplified from the pcDNA3.1-GFP-MDC1 plasmid[11] and cloned into the pEGFP-C1 vector. To create the pEGFP-C1-1X-PST, pEGFP-C1-2X-PST and pEGFP-C1-5X-PST plasmids the truncated PST fragments were amplified from the pEGFP-C1-PST vector using the primers EGFP-C, PST_r2_R and PST_r6_R (Supplementary Table 2).

Bands of the correct sizes were cut from the gel, purified, digested with SmaI and BsrGI, and cloned into pEGFP-C1. All the constructs were verified by Sanger sequencing.

**siRNA and plasmid transfection**. Plasmid transfections were carried out using TransIT-LT1 (Mirus Bio) according to the manufacturer's protocol. RPE-1 cells were transfected with siRNAs obtained from MWG using Lipofectamine RNAi-MAX (Invitrogen) following the manufacturer's instruction. The siRNA final concentration was 60 nM. The sequences of siRNAs (ThermoFisher) are listed in Supplementary Table 3.

**DNA-damage induction**. For induction of 53BP1 nuclear bodies, aphidicolin (Sigma-Aldrich) 0.4 μM was added for 24 h. Ionising radiation treatments at the

indicated doses were performed with a Faxitron-CellRad (Faxitron Bioptics, LLC) machine. In the case of the FRT-derived cells, expression of all the GFP-tagged constructs was induced two days before the start of any treatment (APH or IR) by addition of doxycycline to a final concentration of 0.5 μg ml[−1]. The cells were maintained in the presence of doxycycline for the whole duration of the experiment.

**Clonogenic survival assays**. The day before treatment, cells were seeded in 6-well plates at 500 or 1000 cells per well and three replicates per condition. For the cell lines where a poor survival was expected, 5000 cells were seeded for the highest dose. Upon treatment with IR at the appropriate dose, cells were incubated for 7–10 days, stained with crystal violet, and the number of colonies per well was counted and normalised to the initial number of cells. For all experiments, data were normalised to the untreated conditions to consider variations in plating efficiency. Only colonies with more than 30 cells were considered as proper colonies and therefore counted.

**Whole-cell extracts and immunoblotting**. Whole cell lysates were prepared by scraping cells in 2xSDS buffer (120 mM Tris-HCl pH 6.8, 4% SDS, 20% glycerol). Protein concentration was determined with Nanodrop One (Thermo Scientific) and 30 μg run in 4–12% Bis-Tris NuPAGE precast gels. Separated proteins were transferred to nitrocellulose (GE Healthcare) and immunoblotted with the indicated antibodies. A list of all antibodies used in this study can be found in Supplementary Table 4. All uncropped images are provided in Supplementary Fig. 6a, b.

**Immunoprecipitation**. *MDC1*[−/−] *H2AX*[−/−] U2OS cells were transfected with GFP-PST or GFP-only expressing plasmids and pull-down experiments were carried out two days later. Immunoprecipitation was performed as follows: cells were lysed in benzonase buffer (20 mM Tris-HCl pH 7.5, 40 mM NaCl, 2 mM MgCl$_2$, 10% Glycerol, 0.5% Igepal, EDTA-free protease inhibitors (Roche, 1 tablet per 50 ml), EGTA-free phosphatase inhibitors (1 mM NaF, 0.7 mM β-glycerol phosphate, 0.2 mM Na$_3$VO$_4$, 8.4 mM Na$_4$P$_2$O$_7$), benzonase (Novagen, 3 μl per ml), 3 mM 1,10-phenanthroline, 10 mM N-ethylmaleimide) for 15 min at room temperature. The NaCl concentration was then increased to 500 mM prior to incubating on ice for 20 min. The lysate was then centrifuged at 20000 g for an hour at 4 °C and the supernatant was transferred to a new tube. Salt concentration was adjusted to 150 mM with benzonase buffer and protein concentrations were determined by Bradford assay and adjusted to the lowest concentrated sample with buffer. The protein extracts were then mixed with previously washed and equilibrated GFP-Trap beads (Chromotek, 10 μl per milligram of protein) and rotated overnight at 4 °C. Next day, beads were centrifuged for 2 min at 4 °C and washed with benzonase buffer (without benzonase and containing a final concentration of 200 mM NaCl). After five more washes with the same buffer, proteins were eluted from the GFP-Trap beads in a 5–10-min incubation step at 95 °C in 1.5 × SDS sample buffer and immunoblotted as described in the respective section.

**DNA pull-down**. Magnetic streptavidin Dynabeads (Dynal, M-280; 50 μl bead-slurry per reaction) were washed twice with 2 × Binding and Washing buffer (2 × B&W buffer; 10 mM Tris-HCl pH 7.4 and 2 M NaCl) and subsequently rotated for 15 min at room temperature in the presence of 5 pmol of 5′-biotinylated oligonucleotides of the sequence 5′-Biotin-GCCTACCGGTTCGCGAACCGGTAGGC3′. After washing three times with 1 × B&W, DNA-bound beads were incubated overnight at 4 °C with whole cell lysate prepared as follows: HEK293 cells transfected with either GFP-PST or GFP-only expressing plasmids were lysed in benzonase buffer (the same as used for immunoprecipitations) for 15 min at room temperature. The NaCl concentration was then adjusted to 150 mM and EDTA was added to a final concentration of 10 mM to inactivate the benzonase prior to incubating on ice for 20 min. The lysate was then centrifuged at 20,000 × g for an hour at 4 °C and the supernatant was then mixed with the DNA-coated beads. After binding, the beads were washed five times with the same buffer (without benzonase) and boiled in 1.5 × Laemmli buffer with ß-mercaptoethanol and bromophenol blue to release the proteins from the beads.

**GFP-PST purification and nucleosome pull-down**. HEK293 cells transfected with either GFP-PST or GFP-only expressing plasmids were lysed in benzonase buffer (the same as used for immunoprecipitations) for 15 min at room temperature. Subsequently, the NaCl and Igepal concentrations were increased to 1 M and 1% respectively and samples incubated for 20 min on ice. After centrifugation for 1 h at 20,000 × g at 4 °C, the supernatant was transferred to new tubes and mixed with previously washed and equilibrated GFP-Trap beads (Chromotek). Lysates and beads were rotated at room temperature for 1 h and then washed ten times with the high salt and detergent benzonase buffer (without benzonase). The beads were then washed three more times with binding buffer (10 mM Tris pH7.5, 150 mM NaCl, 0.5% NP40, 0.5 mM EDTA) and suspended in 0.5 ml of the same buffer. In all, 5 μg of nucleosome (Epicypher) or histone mix (Sigma-Aldrich) were added to the beads and rotated overnight at 4 °C. Next morning the beads were washed five times with binding buffer and proteins were eluted by boiling in Laemmli buffer with ß-mercaptoethanol and bromophenol blue.

**Silver staining**. Silver staining was performed with the SilverQuest™ Silver Staining Kit (Invitrogen) following the manufacturer's recommendations.

**Immunofluorescence staining**. Cells were grown on poly-lysine-coated coverslips for treatments with IR or aphidicolin. Following treatment, cells were washed with PBS 0.1% Tween and fixed without pre-extraction with 2% paraformaldehyde in PBS for 20 min at room temperature. Samples were then permeabilised with PBS 0.5% Triton for 15 min, blocked in 5% bovine serum albumin in PBS and stained with the appropriate primary antibody (Supplementary Table 4) and secondary antibodies coupled to Alexa Fluor 488, 594 or 647 (Molecular Probes). When samples were subjected to pre-extraction, coverslips were incubated with pre-extraction buffer (25 mM HEPES pH 7.4, 50 mM NaCl, 1 mM EDTA, 3 mM MgCl$_2$, 300 mM sucrose and 0.5% Triton X-100) for 15 min on ice and then washed with PBS and fixed as above. Pre-extracted samples did not require further permeabilization, so that step was omitted. Confocal images were captured on a Leica SP8 confocal microscope with a ×40 or ×60 oil objective lens and processed by ImageJ (version 2.0.0) and quantified by Volocity 6.3 (PerkinElmer).

**Automated high-throughput/high-content microscopy**. For nuclear bodies and IRIF quantifications, cells were seeded into 96-well plates (Cell Carrier, Perkin Elmer) at a density of 10,000 cells per well. The following day, cells were either mock treated, treated with aphidicolin or irradiated (3 Gy). Plates were fixed at the indicated time points and stained with the respective antibodies and DAPI. A spinning-disc Perkin Elmer Opera platform equipped with a ×40 water immersion objective was employed to acquire 10 confocal images (fields) for each well in a single optimised focal plane comprising two fluorescence channels, DAPI and Alexa Fluor 488. The micrographs were analysed using an optimised spot detection script operated by an integrated software package (Harmony 4.8, Perkin Elmer). DAPI was used to segment nuclei and create a nuclear mask. In the case of nuclear body quantification, cyclin A positive cells were excluded and the percentage of 53BP1-IRIF positive or 53BP1-NBs positive cells calculated. For FRT cell lines expressing the GFP-MDC1 constructs, only GFP-positive cells were taking into account in the final population. More than 500 cells were counted per condition and repeat.

**Cell cycle profiling**. Cells were collected after a 30 min pulse with 10 μM EdU, washed with ice-cold PBS and fixed for 30 min with 2% paraformaldehyde in PBS. After washing with PBS-B (1xPBS + 1 mg ml$^{-1}$ BSA), cells were permeabilized with PBS-T (1 × PBS + 0.2% Triton X-100) and incubated with the Click reaction cocktail (2 mM CuSO$_4$, 1 μM Alexa Fluor™ 488 Azide (Invitrogen), 10 mM sodium L-ascorbate, in PBS) for 30 min at room temperature in the dark. Finally, cells were washed with PBS-B and suspended in FACS buffer (0.02% NaAz, 250 μg ml$^{-1}$ RNase, 3.2 ng ml$^{-1}$ DAPI, in PBS-B). Flow cytometry was performed with an Attune NxT machine (Invitrogen) and analysed with FlowJo software (BD Inc, USA).

**Chromatin fractionation**. Cells were collected from plates, washed with cold PBS and suspended in CSK buffer (10 mM PIPES pH 7.0, 100 mM NaCl, 300 mM sucrose, 3 mM MgCl$_2$, protein inhibitor cocktail (Roche, EDTA-free), phosphatase inhibitors (EGTA-free, same as for benzonase buffer), 0.7% Triton X-100). After 30 min incubation on ice, the lysate was centrifuged at 20,000 × g for 10 min at 4 °C. The supernatant (soluble fraction) was collected and kept on ice. The pellet was washed with cold PBS, suspended in CSK buffer and sonicated for four pulses of 10 s at 30% amplitude with 10 s resting on ice between cycles. This sonicated solution is the chromatin fraction. Laemmli buffer was added to the soluble and the chromatin fractions and both samples were boiled and centrifuged for 1 min at 16,000 × g in a table-top centrifuge. In total, 30–50 μg of total protein from each fraction were loaded onto SDS-PAGE gels.

**Recombinant nucleosome generation**. Defined recombinant nucleosome core particles (NCPs) were generated essentially as described[17]. Histone variants were mutated by site-directed mutagenesis to introduce acidic patch mutations and remove cysteines in histone H3. Briefly, individual core human histones and histone variants (H2A.1, H2A.1 e65A E90A E91A; H2B.1, H3.1 C110A, C96S, R134C, H4) were expressed in BL-21 DE3 RIL E. coli and histones were purified from inclusion bodies via cation exchange chromatography[55], dialysed into water supplemented with 1 mM acetic acid and lyophilised for storage at −20 °C. Pure histones were mixed at an equimolar ratio and dialysed into a high salt buffer (2 M NaCl, 15 mM Tris pH 7.5, 1 mM DTT). Octamers containing histones were selected by size exclusion chromatography (HiLoad HiLoad Superdex 200 16/600 GE healthcare S200) and concentrated in 30 kDa MWCO spin concentrator (Amicon). 169 bp of Widom-603 strong nuclear positioning DNA was generated by PCR using Pfu polymerase and HPLC-grade oligonucleotides essentially as described[56]. In all, 100 μl PCR products from 384 reactions were pooled and purified on a ResorceQ column (GE Healthcare) and 169 bp product fractions concentrated by ethanol precipitation. NCPs were reconstituted by slow gradient dialysis of equimolar DNA and histone octamers into low salt buffer[55] (15 mM HEPEs pH 7.5, 200 mM KCl, 1 mM DTT, 1 mM EDTA), prior to a final dialysis in storage buffer (15 mM HEPEs pH 7.5, 100 mM NaCl, 1 mM EDTA, 1 mM DTT). Soluble NCPs were concentrated using a 100-kDa cut-off spin concentrator (Amicon). NCP formation and purity were confirmed by native polyacrylamide gel electrophoresis and NCPs were stored at 4 °C and used within one month of assembly.

**Statistics**. Statistical analysis (Two-tailed Student's t tests) in all graphs of this work were calculated using GraphPad Prism version 8 for Mac OS X, GraphPad Software (La Jolla, CA, USA).

**Reporting summary**. Further information on research design is available in the Nature Research Reporting Summary linked to this article.

## Data availability
All relevant data are available from the authors upon request. The source data underlying. Figs. 1b, 2b, d, 3b–f, and 4c are provided as a Source Data file.

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

## Acknowledgements

We thank all S.P.J. laboratory members and Dr. Andrew N. Blackford for discussions, and Dr. Kate Dry for editorial assistance. Research in the S.P.J. laboratory is funded by Cancer Research UK (programme grant C6/A18796) and Wellcome Investigator Award (206388/Z/17/Z). Institute core infrastructure funding is provided by Cancer Research UK (C6946/A24843) and Wellcome (WT203015). S.P.J. receives salary from the University of Cambridge. This work was funded by Cancer Research UK programme grant C6/A18796 (I.S., R.B., J.C., M.S-C. and M.D.), Wellcome Strategic Award 101126/Z/13/Z (I.S.) and ERC Advanced Research Grant DDREAM (M.S-C., M.D. and S.J.). M.D.W's work is supported by Wellcome (210493) and the University of Edinburgh. The Wellcome Centre for Cell Biology is supported by core funding from Wellcome [203149]. Histone plasmids were a gift from Joe Landry (Addgene).

## Author contributions

I.S. and S.P.J. largely conceived the study and wrote the manuscript. I.S. assembled the figures and supervised/was involved in all the experiments. R.B. generated the plasmids for expression of the different truncated versions of GFP-PST, helped with the design of the in vitro nucleosome pull-down experiments and the high-throughput microscopy, and contributed with extensive discussions throughout the work. J.C. helped with the clonogenic experiments, immunofluorescence microscopy and with the generation of the FRT-derived cells stably expressing inducible GFP-tagged constructs. M.S-C. and M.D. created the U2OS *H2AX$^{-/-}$* single knockout and all the *H2AX$^{-/-}$ MDC1$^{-/-}$* double knockout cell lines. M.S-C. also genotyped the single and double knockout cell lines. S.J. generated the RPE-1 *H2AX$^{-/-}$* cell line. M.D.W. produced the reconstituted recombinant wild-type and acidic patch mutant nucleosomes and advised on the in vitro GFP-PST/nucleosome-binding experiments design. All authors commented on the manuscript and figures. S.P.J. supervised the work.

## Competing interests

The authors declare no competing interests.
