## [Peer Review File · Nature Communications]

Reviewers' comments:

Reviewer #1 (Remarks to the Author):

In this manuscript, Salguero et al. use CRISPR/Cas9-engineered cell line deficient in H2AX, MDC1, or both to shed new light on an old problem: Phosphorylated H2AX (the so-called γ H2AX) is one of the earliest histone modifications, which marks the physical sites of DNA breaks. As this laboratory previously discovered, the phosphate on the C-terminus of γ H2AX is recognized by MDC1, which extends the chromatin modifications to distant chromatin territories and thus amplifies the DNA damage signaling. This model has served as a fundament for much of the DNA repair field but since its early days, there have been two puzzling observations that complicate the picture. First, MDC1 knockout mice have more severe phenotypes compared to their H2AX knockout counterparts. Second, it was observed that some of the downstream repair regulators transiently migrate to sites of DNA breakage that have not been marked by γ H2AX. Here, the authors address both questions by showing that MDC1 can bind nucleosome acidic patches on chromatin independent of γ H2AX by a mechanism utilizing a conserved PST-repeat region. They provide data suggesting that the PST-mediated interaction of MDC1 with chromatin is sufficient to support DNA damage signaling, especially in chromatin regions that are naturally low in H2AX histone variant.

As such, this work is topical, the questions are interesting, and their clarification would be useful for the field to interpret old and new data dealing with amplification of DNA damage signaling from a broader perspective. However, I have a number of problems with the quality of the data and their interpretation. I will first outline what I think should be done to improve the technical quality. I will then summarise specific concerns to individual figures. In the last part, I will comment on the general mechanistic advancement of this work.

Technical or formal shortcomings in the text that need to be addressed:

1. Neither figure legends, nor figure layouts specify the name/type of the cell lines used.
2. siRNA concentrations are not indicated in the methods.
3. Although the methods section states that: "FRT-derived cells stably expressing inducible GFP-tagged constructs", neither legends, nor figure labeling indicate whether, how long and under which conditions were the transgenes induced.
4. Experimental treatments (IR, APH) are only sporadically and not consistently specified in figure legends or layouts.
5. The conditions for survival assays need to be extended so that the reader can judge the strength of these data (number of cells seeded, definition of a colony to be counted as survival, etc.); it is not sufficient to refer to previous work that seem to have been performed with a different cell type.
6. The most significant shortcoming is the lack of proper characterization of the knockout cell lines. Western blotting is not sufficient to claim full knockout of all alleles and such level of analysis cannot exclude residual amounts of the targeted factor that can have residual function. Genomic data are needed to validate successful gene targeting.
7. Related to the previous criticism, the cell cycle analysis of the knockout clones needs to be provided; without this information, it is difficult to interpret the data.

Specific comments to figures:

8. Fig. 2b (the same criticism applies to Fig. 3d, and Supplementary Fig. 2a; all experiments that quantitate nuclear foci): The cutoff "Cells with > 3 IRIF" is arbitrary and outdated in the current era of quantitative microscopy. It seems that the authors have interesting single-cell data for number of IRIF and NBs. Rather than using this rough "more than three" measure to represent DNA damage responses (which can discard groups or subgroups of interesting phenotypes), it would be much more informative and convincing to plot the number of IRIF and NBs for single cells in a dot-plot.
9. Fig. 4a and the related Supplementary Fig. 4a: These data are difficult to interpret because the GFP-PST fragment is clearly sequestered in nucleoli (clearly evident on images in Supplementary

Fig. 4a but not mentioned in the manuscript). This subcellular sequestration raises the question what is really measured as a “chromatin fraction”.

10. Another comment to Supplementary Fig. 4a: the images lack information about cyclin A; it is therefore impossible to know whether the depicted cells treated by APH are in G1 to support the expected readout.

11. Fig. 4c and the related Supplementary Fig. 4c: The quality of these western blots is poor and the conclusions derived from these data are unconvincing. The authors fail to provide quantitative in vivo evidence for different chromatin affinity of MDC1 and its PST-deficient version, respectively. In addition, lane 1 in Fig. 4a contains a clear band migrating with a size of MDC1 Δ PST, while it should only contain full-length MDC1.

12. Fig. 4f: Similar problems with the quality as in the previous point: the bands are weak and quantitative conclusion about DNA binding cannot be derived from these data.

General comments on mechanistic advancement:

13. I was missing an explanation of how can a protein (MDC1) proposed to bind chromatin in a sequence/locus-independent manner support local reactions at the sites of DNA breaks? If H2AX is deleted, what restrains MDC1 function to the breakage site?

14. The current model posits that H2AX phosphorylation serves to amplify DNA damage signaling. I was interested to know whether the authors want to challenge this model? It is interesting but also surprising to see that DNA damage signaling (Supplementary Fig. 1c and 5) does not seem to show any difference in H2AX or MDC1 deficient genetic backgrounds.

Reviewer #2 (Remarks to the Author):

In this manuscript, Salguero and colleagues identify a role for the PST repeats of MDC1 in the H2AX-independent response to DNA double-strand breaks. They first show that MDC1 loss results in a stronger sensitization to ionizing radiation than H2AX loss in isogenic engineered cell lines. They then observe that the loss of MDC1 is particularly deleterious for the accrual of 53BP1 at G1 nuclear bodies. They then carry out a structure-function study and found that the PST repeat region of MDC1 is necessary to promote the localization of MDC1 at G1-NBs, and that of 53BP1 at G1-NBs in the absence of H2AX. The PST repeat region also promotes resistance to IR in the absence of H2AX. Finally, the authors show that the PST repeat region interacts with the acidic patch of the nucleosome, providing a plausible mechanism for the H2AX-independent retention of MDC1 on damaged chromatin.

Overall, this manuscript provides a very useful advance to the field as it reveals molecular determinants of H2AX-dependent signaling at DNA double-strand break sites. I am supportive about this well executed work but I hope the authors can clarify a few of the points elaborated below.

Specific points:

1) The distinct genetic dependencies between G1-NBs and IRIFs are striking. One possible source of these differences could be related to the observation that MDC1 plays additional roles at mitotic DSBs (Leimbacher et al. 2019). This might also be consistent with other data, such as the data shown in Fig 3f. Does the PST repeat region help MDC1 form mitotic IRIFs independently of H2AX?

2) One unresolved question is whether the binding of the PST region to the acidic patch is in fact necessary for MDC1 function in the absence of H2AX. The manuscript would be stronger if they could identify a mutation (or set of mutations) in the PST region that disrupts nucleosome-binding and then assess functional impact of the mutation.

3) In the same vein, does a single PST repeat interact with the nucleosomal acidic patch? The nucleosome-binding motif in the PST region could well be a sequence other than the PST motif.

4) The RNF8-53BP1-dependent recruitment of MDC1 in G1-NBs (fig 2c-d) is really intriguing. Is it also dependent on shieldin?

Minor point

- On p4, the authors seem to make a distinction between REV7 and shieldin (in the first paragraph) but REV7 is an integral part of shieldin so that sentence could be revised.

Reviewers' comments:

Reviewer #1 (Remarks to the Author):

In this manuscript, Salguero et al. use CRISPR/Cas9-engineered cell line deficient in H2AX, MDC1, or both to shed new light on an old problem: Phosphorylated H2AX (the so-called γ H2AX) is one of the earliest histone modifications, which marks the physical sites of DNA breaks. As this laboratory previously discovered, the phosphate on the C-terminus of γ H2AX is recognized by MDC1, which extends the chromatin modifications to distant chromatin territories and thus amplifies the DNA damage signaling. This model has served as a fundament for much of the DNA repair field but since its early days, there have been two puzzling observations that complicate the picture. First, MDC1 knockout mice have more severe phenotypes compared to their H2AX knockout counterparts. Second, it was observed that some of the downstream repair regulators transiently migrate to sites of DNA breakage that have not been marked by γ H2AX. Here, the authors address both questions by showing that MDC1 can bind nucleosome acidic patches on chromatin independent of γ H2AX by a mechanism utilizing a conserved PST-repeat region. They provide data suggesting that the PST-mediated interaction of MDC1 with chromatin is sufficient to support DNA damage signaling, especially in chromatin regions that are naturally low in H2AX histone variant.

As such, this work is topical, the questions are interesting, and their clarification would be useful for the field to interpret old and new data dealing with amplification of DNA damage signaling from a broader perspective. However, I have a number of problems with the quality of the data and their interpretation. I will first outline what I think should be done to improve the technical quality. I will then summarise specific concerns to individual figures. In the last part, I will comment on the general mechanistic advancement of this work.

We thank this reviewer for his/her positive assessment of our work, and for his/her important and insightful comments and suggestions.

Technical or formal shortcomings in the text that need to be addressed:

1. Neither figure legends, nor figure layouts specify the name/type of the cell lines used.

We apologize for this. The cell lines used for each experiment are specified in the figure legends of the new version of the manuscript.

2. siRNA concentrations are not indicated in the methods.

This has been corrected in the new version of the manuscript.

3. Although the methods section states that: "FRT-derived cells stably expressing inducible GFP-tagged constructs", neither legends, nor figure labeling indicate whether, how long and under which conditions were the transgenes induced.

The reviewer is correct to point out these omissions. Expression of all the GFP-tagged constructs in the FRT-derived cells was induced two days before the start of any treatment (APH or IR) by addition of doxycycline to a final concentration of 0.5 μ g/ml. The cells were maintained in the presence of doxycycline for the whole duration of the experiment. This information has now been added to the revised Methods section (DNA-damage induction).

4. Experimental treatments (IR, APH) are only sporadically and not consistently specified in figure legends or layouts.

These are now indicated in a clearer way in the figure layouts and more detailed ways in the figure legends.

5. The conditions for survival assays need to be extended so that the reader can judge the strength of these data (number of cells seeded, definition of a colony to be counted as survival, etc.); it is not sufficient to refer to previous work that seem to have been performed with a different cell type.

A more detailed description of the assay is now included in the Methods section.

6. The most significant shortcoming is the lack of proper characterization of the knockout cell lines. Western blotting is not sufficient to claim full knockout of all alleles and such level of analysis cannot exclude residual amounts of the targeted factor that can have residual function. Genomic data are needed to validate successful

gene targeting.

7. Related to the previous criticism, the cell cycle analysis of the knockout clones needs to be provided; without this information, it is difficult to interpret the data.

We thank this reviewer for pointing out that the knockout cell lines generated in this work needed further characterisation (Points 6 and 7). We have now included genotypic information, describing the mutations generated by the CRISPR/Cas9 system, and cell cycle profile for each of the cell lines (Supplementary Figures 1b, c, and 4c, d).

Specific comments to figures:

8. Fig. 2b (the same criticism applies to Fig. 3d, and Supplementary Fig. 2a; all experiments that quantitate nuclear foci): The cutoff "Cells with > 3 IRIF" is arbitrary and outdated in the current era of quantitative microscopy. It seems that the authors have interesting single-cell data for number of IRIF and NBs. Rather than using this rough "more than three" measure to represent DNA damage responses (which can discard groups or subgroups of interesting phenotypes), it would be much more informative and convincing to plot the number of IRIF and NBs for single cells in a dot-plot.

The numbers of IRIF and NBs for single cells have now been included. The cutoff "cells with >3 IRIFs" was set to discriminate proper IRIF-positive cells from the background noise due to protein aggregation, artefacts from staining or even the nuclear bodies present in the cell population. In addition, due to the reduced population of IRIF-positive cells in the H2AX null background, we believe that the percentage of positive cells, rather than the number of IRIFs per cell, would be more interesting for the readers. Therefore, we have included the number of foci per cell quantifications in the Supplementary information (Supplementary Figures 2a and 3b, e).

9. Fig. 4a and the related Supplementary Fig. 4a: These data are difficult to interpret because the GFP-PST fragment is clearly sequestered in nucleoli (clearly evident on images in Supplementary Fig. 4a but not mentioned in the manuscript). This subcellular sequestration raises the question what is really measured as a "chromatin fraction".

We thank the reviewer for raising this issue. We have now included a figure with immunofluorescence images showing that its association with nucleoli is lost upon pre-extraction conditions. Taking into account that detergent buffers used for pre-extraction in immunofluorescence and for chromatin fractionation are very similar in composition, these observations, in our opinion, strongly argue against the possibility of the chromatin retention of the PST fragment being mediated by its enrichment in nucleoli (Supplementary Figure 4e).

10. Another comment to Supplementary Fig. 4a: the images lack information about cyclin A; it is therefore impossible to know whether the depicted cells treated by APH are in G1 to support the expected readout.

We have repeated this experiment co-staining for Cyclin A for better discrimination of nuclear bodies (Supplementary Figure 4a).

11. Fig. 4c and the related Supplementary Fig. 4c: The quality of these western blots is poor and the conclusions derived from these data are unconvincing. The authors fail to provide quantitative *in vivo* evidence for different chromatin affinity of MDC1 and its PST-deficient version, respectively. In addition, lane 1 in Fig. 4a contains a clear band migrating with a size of MDC1 Δ PST, while it should only contain full-length MDC1.

These experiments have been repeated to obtain better quality images, and quantifications have been included (Fig. 4c). Regarding the lower molecular weight band in the full-length MDC1 lane, it is well reported in the literature that MDC1 migrates as two or three distinct bands in western blot studies (e.g. Stewart *et al.*, 2003; Goldberg *et al.*, 2003; Stucki *et al.*, 2005). It is not clear whether these bands correspond to degradation products, to differential posttranslational modifications or both. In any case, although that band migrates very close to the lowest migrating band of full-length MDC1, they can be separated by running the gel for longer. Therefore, if the reviewer is still not convinced, we would be happy to show an image corresponding to a longer-run western.

12. Fig. 4f: Similar problems with the quality as in the previous point: the bands are weak and quantitative conclusion about DNA binding cannot be derived from these data.

As the reviewer mentions DNA binding, we assume he/she means Supplementary Fig. 4f. We believe there has been a misunderstanding with this panel. The only weak bands in this western are the ones in the overexposed film that we are including just to show that there is some unspecific binding to the beads, but this is the case both

for the GFP-PST and the GFP-only constructs. Therefore, this result suggests that a direct binding of the PST repeats to DNA is not mediating the PST association to nucleosomes.

General comments on mechanistic advancement:

13. I was missing an explanation of how can a protein (MDC1) proposed to bind chromatin in a sequence/locus-independent manner support local reactions at the sites of DNA breaks? If *H2AX* is deleted, what restrains MDC1 function to the breakage site?

We thank the reviewer for this comment. In our opinion, ATM would be activated at the break by the MRN complex and associated factors, in the same way as in the case of having H2AX... and would thus act locally, only phosphorylating MDC1 molecules in the vicinity of the break. We make this point in our revised text at the end of the Discussion section.

14. The current model posits that H2AX phosphorylation serves to amplify DNA damage signaling. I was interested to know whether the authors want to challenge this model? It is interesting but also surprising to see that DNA damage signaling (Supplementary Fig. 1c and 5) does not seem to show any difference in H2AX or MDC1 deficient genetic backgrounds.

We did not expect to find a big signalling defect in the H2AX knockout, as MDC1 would still contribute to the signalling cascade, but it was surprising to us to observe no apparent signalling defects in the MDC1 knockout background either. It is true that ATM is in the first instance recruited to the break and activated by the MRN complex. In the same way, resection will also happen at some breaks independently of H2AX and MDC1, and this will activate ATR, increasing the signalling reactions. Nevertheless, it is still unexpected to see no defects in the absence of MDC1, but the same phenomenon has been recently reported by others for MDC1 knockouts in U2OS cells (Leimbacher *et al.*, 2019). Early literature on MDC1 report contradictory effects of MDC1 depletion on the signalling pathway, with some papers describing reduced phosphorylation of some DDR factors and other papers showing a bigger effect on other phosphorylations (e.g. Stewart *et al.*, 2003; Goldberg *et al.*, 2003; Minter-Dykhouse *et al.*, 2008; Townsend *et al.*, 2009). One explanation for the discrepancies between our observations and previous ones could be off-target effects of siRNAs used in the previous studies and/or the adaptation experienced by our cell lines after knocking out *MDC1*. We now briefly cover these issues in the Discussion section of our revised manuscript.

Reviewer #2 (Remarks to the Author):

In this manuscript, Salguero and colleagues identify a role for the PST repeats of MDC1 in the H2AX-independent response to DNA double-strand breaks. They first show that MDC1 loss results in a stronger sensitization to ionizing radiation than H2AX loss in isogenic engineered cell lines. They then observe that the loss of MDC1 is particularly deleterious for the accrual of 53BP1 at G1 nuclear bodies. They then carry out a structure-function study and found that the PST repeat region of MDC1 is necessary to promote the localization of MDC1 at G1-NBs, and that of 53BP1 at G1-NBs in the absence of H2AX. The PST repeat region also promotes resistance to IR in the absence of H2AX. Finally, the authors show that the PST repeat region interacts with the acidic patch of the nucleosome, providing a plausible mechanism for the H2AX-independent retention of MDC1 on damaged chromatin.

Overall, this manuscript provides a very useful advance to the field as it reveals molecular determinants of H2AX-dependent signaling at DNA double-strand break sites. I am supportive about this well executed work, but I hope the authors can clarify a few of the points elaborated below.

We are very thankful to this reviewer for his/her positive evaluation of our work and for his/her interesting and constructive comments.

Specific points:

1) The distinct genetic dependencies between G1-NBs and IRIFs are striking. One possible source of these differences could be related to the observation that MDC1 plays additional roles at mitotic DSBs (Leimbacher *et al.*, 2019). This might also be consistent with other data, such as the data shown in Fig 3f. Does the PST repeat region help MDC1 form mitotic IRIFs independently of H2AX?

This is a very interesting possibility that we had not taken into account. To test this hypothesis, the *MDC1* KO and the *MDC1/H2AX* KO RPE-I cell lines complemented with either full-length or Δ PST GFP-MDC1 were arrested in mitosis by nocodazole. The cells were then irradiated with 0.5 Gy and fixed 1 hour later. We could detect MDC1 IRIF formation and colocalization with TOPBP1 in *MDC1* KO cells expressing the Δ PST version of

GFP-MDC1. To our surprise, we could also detect MDC1 foci formation in the *MDC1/H2AX* double knockout, although these foci were less frequent and smaller than in the *H2AX^{+/+}* cells (see below). In addition, these MDC1 IRIF did not seem to depend on the PST repeats. Apart from NBs, this is the first time we have been able to detect MDC1 enrichment at DNA-damage sites in the absence of H2AX. We do not have an explanation for this observation, but the fact that it does not happen in interphase cells and that it does not depend on the PST repeats, even in backgrounds lacking H2AX, suggests that this is an alternative mechanism to recruit MDC1 that is specific to mitosis. Maybe, the DNA breaks in mitosis expose regions of chromatin that otherwise would be extremely compacted and inaccessible to MDC1. It will be interesting to find out which region(s) of MDC1 is/are responsible for this localization and it is an issue that we intend to address in the future. However, we feel that clarification of this behaviour is out of the scope of our current manuscript. We point out that we have now included a section on MDC1's mitotic DDR roles in the introduction section of our revised manuscript.

2) One unresolved question is whether the binding of the PST region to the acidic patch is in fact necessary for MDC1 function in the absence of H2AX. The manuscript would be stronger if they could identify a mutation (or set of mutations) in the PST region that disrupts nucleosome-binding and then assess functional impact of the mutation.

3) In the same vein, does a single PST repeat interact with the nucleosomal acidic patch? The nucleosome-binding motif in the PST region could well be a sequence other than the PST motif.

To address points 2 and 3, we generated and purified MDC1 derivatives containing 1, 2 or 5 PST-region repeats fused to GFP and tested their ability to bind nucleosomes *in vitro*. We were able to detect binding to nucleosomes only with the PST fragment containing 5 repeats, and even this association was substantially reduced when compared with the full-length PST fragment. We then examined the chromatin binding capacity of these constructs in cells and found that, in agreement with the *in vitro* pull-downs, only the 5-PST and full-length proteins were enriched in the chromatin fraction, with the full-length construct being substantially more associated with chromatin than the 5-PST version.

In our opinion these data are interesting, and support a model in which different PST repeats inside the same MDC1 molecule interact with different acidic patches belonging to the same or different nucleosomes. This

cooperative binding mode might thus contribute to the strength and/or stability of the interaction, and would help explain the fact that MDC1 proteins from diverse species have multiple PST-repeats. We have included these new results in the last figure of the paper (Fig. 5c, d), and discuss them in our revised text.

Unfortunately, the requirement for multiple PST motifs to mediate chromatin/nucleosome binding will make it difficult to define key functional residues: to identify the key residues in the repeats that mediate the interaction with the acidic patch, we would have to mutate different candidate amino acid residues in each one of the 13 repeats of the full-length PST region (or in the 5x PST-repeat construct) and then carry out binding studies. Furthermore, if such work were successful in finding the key residues, we would then have to stably introduce these and control constructs into cells of the *MDC* null genetic background... and then evaluate their potential effects on the various cell physiology readouts used in this work. Unfortunately, to carry all these experiments out would be impossible in the 3-month timeframe available. In light of this, we hope that this reviewer and the Editor will agree that such analyses are outside the scope of the current manuscript.

4) The RNF8-53BP1-dependent recruitment of MDC1 in G1-NBs (fig 2c-d) is really intriguing. Is it also dependent on shieldin?

We have addressed this point by depleting SHLD1 or SHLD2 in *H2AX*^{+/+} and *H2AX*^{-/-} RPE-1 cells, and then quantifying the frequencies of MDC1-NBs positive cells among the G1 populations, in the same way we did for 53BP1 and RNF8. These results have been included in Figure 2d and Supplementary figure 2c and d. The answer to the reviewer's question is that the recruitment and/or retention of MDC1 at NBs in the *H2AX*^{-/-} mutant cells does not depend on SHLD1/2 complex. This supports the hypothesis of a direct interaction between 53BP1 and MDC1 being necessary for MDC1 enrichment in NBs.

Minor point

On p4, the authors seem to make a distinction between REV7 and shieldin (in the first paragraph) but REV7 is an integral part of shieldin so that sentence could be revised.

This has been corrected in the manuscript.

REVIEWERS' COMMENTS:

Reviewer #1 (Remarks to the Author):

This is a revised version of the manuscript by Salguero et al., where the authors came a long way to address my previous criticisms and also those made by the other referees.

In general, I am satisfied with the answers to my technical and formal shortcomings – there is only one exception that might have been caused by formatting the files for submission: All the authors' answers that relate to Supplementary Fig. 4 (my previous points 7, 9, 10 and 12) are well described in the rebuttal but difficult to judge, because this Figure lacks panels a-e; in all related files I could upload, Supplementary Fig. 4 always starts with panel f. This needs to be corrected.

The answers to the more conceptual issues (previous points 13 and 14) are only speculative and not entirely satisfactory, but I appreciate the authors' effort to comment on these issues and also reflect these in the revised manuscript. On the other hand, I like the new additions made in response to the referee #2; the new data suggesting cooperative binding of the PST repeats as a means to stabilize MDC1 interaction with nucleosome acidic patches (Fig. 5c, d) are interesting and shed some light on structural features of these domains in diverse species. This indeed strengthens the mechanistic part of the paper.

On aggregate, the revised manuscript has improved and it does provide an important contribution to understand a long-standing puzzle in the field. I therefore recommend publication with the correction of the Supplementary Fig. 4 as detailed above.

Reviewer #2 (Remarks to the Author):

The authors have addressed my comments adequately. Thank you.

REVIEWERS' COMMENTS:

Reviewer #1 (Remarks to the Author):

This is a revised version of the manuscript by Salguero et al., where the authors came a long way to address my previous criticisms and also those made by the other referees.

In general, I am satisfied with the answers to my technical and formal shortcomings – there is only one exception that might have been caused by formatting the files for submission: All the authors' answers that relate to Supplementary Fig. 4 (my previous points 7, 9, 10 and 12) are well described in the rebuttal but difficult to judge, because this Figure lacks panels a-e; in all related files I could upload, Supplementary Fig. 4 always starts with panel f. This needs to be corrected.

The answers to the more conceptual issues (previous points 13 and 14) are only speculative and not entirely satisfactory, but I appreciate the authors' effort to comment on these issues and also reflect these in the revised manuscript. On the other hand, I like the new additions made in response to the referee #2; the new data suggesting cooperative binding of the PST repeats as a means to stabilize MDC1 interaction with nucleosome acidic patches (Fig. 5c, d) are interesting and shed some light on structural features of these domains in diverse species. This indeed strengthens the mechanistic part of the paper.

On aggregate, the revised manuscript has improved and it does provide an important contribution to understand a long-standing puzzle in the field. I therefore recommend publication with the correction of the Supplementary Fig. 4 as detailed above.

We thank reviewer 1 for all his/her constructive criticism along the whole review process. We are glad to read that he/she is satisfied with the improvement of the manuscript. Regarding Supplementary Fig. 4, the reviewer is right; it seems we have missed part of that file during the formatting prior the previous resubmission. We apologize for that. This error has been amended in the new Supplementary Material file.

Reviewer #2 (Remarks to the Author):

The authors have addressed my comments adequately. Thank you.

We are very thankful to this reviewer for his/her positive evaluation of our work. We believe the manuscript has improved considerably thanks to the suggestions of this and our other reviewer.